# Identification of Metabolomics Biomarkers in Extracranial Carotid Artery Stenosis

**DOI:** 10.3390/cells11193022

**Published:** 2022-09-27

**Authors:** Chia-Ni Lin, Kai-Cheng Hsu, Kuo-Lun Huang, Wen-Cheng Huang, Yi-Lun Hung, Tsong-Hai Lee

**Affiliations:** 1Department of Laboratory Medicine, Linkou Chang Gung Memorial Hospital, Taoyuan 333, Taiwan; 2Department of Medical Biotechnology and Laboratory Science, Chang Gung University, Taoyuan 333, Taiwan; 3School of Medicine, College of Medicine, Artificial Intelligence Center for Medical Diagnosis, and Department of Neurology, China Medical University Hospital, Taichung 404327, Taiwan; 4Stroke Center and Department of Neurology, Linkou Chang Gung Memorial Hospital, Taoyuan 333, Taiwan; 5College of Medicine, Chang Gung University, Taoyuan 333, Taiwan; 6Department of Nuclear Medicine, Linkou Chang Gung Memorial Hospital, Taoyuan 333, Taiwan

**Keywords:** ischemic stroke, carotid artery stenosis, metabolomics, decision tree, random forest

## Abstract

The biochemical identification of carotid artery stenosis (CAS) is still a challenge. Hence, 349 male subjects (176 normal controls and 173 stroke patients with extracranial CAS ≥ 50% diameter stenosis) were recruited. Blood samples were collected 14 days after stroke onset with no acute illness. Carotid plaque score (≥2, ≥5 and ≥8) was used to define CAS severity. Serum metabolites were analyzed using a targeted Absolute IDQ^®^p180 kit. Results showed hypertension, diabetes, smoking, and alcohol consumption were more common, but levels of diastolic blood pressure, HDL-C, LDL-C, and cholesterol were lower in CAS patients than controls (*p* < 0.05), suggesting intensive medical treatment for CAS. PCA and PLS-DA did not demonstrate clear separation between controls and CAS patients. Decision tree and random forest showed that acylcarnitine species (C4, C14:1, C18), amino acids and biogenic amines (SDMA), and glycerophospholipids (PC aa C36:6, PC ae C34:3) contributed to the prediction of CAS. Metabolite panel analysis showed high specificity (0.923 ± 0.081, 0.906 ± 0.086 and 0.881 ± 0.109) but low sensitivity (0.230 ± 0.166, 0.240 ± 0.176 and 0.271 ± 0.169) in the detection of CAS (≥2, ≥5 and ≥8, respectively). The present study suggests that metabolomics profiles could help in differentiating between controls and CAS patients and in monitoring the progression of CAS.

## 1. Introduction

Stroke is a major cause of functional disability and death worldwide, and the economic expense of treatment and post-stroke care is a great burden to the society [1]. Atherosclerosis is a chronic disease of complex etiology and usually results in arterial stenosis and acute thrombosis. Among ischemic stroke subtypes, our previous study showed that carotid artery stenosis (CAS) carries a high mortality rate, with 23.7% mortality upon recurrent stroke [2]. Recent genome-wide association studies in Caucasians [3,4,5] found six genetic loci, and our study in Han Chinese [6] found two loci that were associated with CAS. Among these loci, histone deacetylase 9-TWIST1 (HDAC9-TWIST1, encoding histone deacetylase 9) on chromosome 7p21.1 was found in both ethnicities. However, due to the heterogeneity of stroke, genetic studies did not consistently find replicable genetic risk factors for cerebral atherosclerosis, which may raise difficulty in stroke prevention for CAS.

Despite the advanced diagnostic tools, such as angiography and ultrasound study, in detecting CAS, the biochemical identification of CAS is still a challenge. Metabolomics is known as a quantitative measurement of the metabolic response of living creatures to pathophysiological stimuli or genetic modification [7]. Metabolomics is used as a promising tool to provide current, single biomarker-based approaches by identifying the global biochemical changes and a good tool in the analysis of disease mechanisms and biomarkers [8]. It is possible that using potent analyses, metabolomics may help in quantifying thousands of different metabolites simultaneously within a given sample to create a great impact on medical practice in CAS.

The use of carotid ultrasound and angiography in the diagnosis of CAS [9] is time-consuming, costly, or even invasive. The application of metabolomics could be of potential as a biomarker to investigate the presence of CAS. Untargeted metabolomics using proton nuclear magnetic resonance (^1^H NMR) spectroscopy has been applied in many fields of atherosclerosis research, including carotid [10,11,12,13,14] and coronary [11,15,16] systems. Our previous study using ^1^H NMR found that homocysteine, choline, and lipids in association with traditional risk factors may be involved in the pathogenesis of extracranial CAS [14]. High intima-media thickness was found best predictable when total and high-density lipoprotein cholesterol (HDL-C) were replaced by NMR-determined low-density lipoprotein cholesterol (LDL-C) and medium HDL-C, docosahexaenoic acid, and tyrosine in combination with the risk factors from Framingham risk score [10]. Atherosclerosis may be associated with metabolites that have disturbances in lipid and carbohydrate metabolism, branched chain, aromatic amino acid metabolism, and also oxidative stress and inflammatory pathways [11]. There are differences in metabolic patterns between extracranial and intracranial carotid artery calcification, with the 3-hydroxybutyrate circulating level being higher in intracranial carotid artery calcification [12].

In the studies using liquid chromatography–mass spectrometry (LC–MS) [17,18,19,20], subjects with asymptomatic severe intracranial artery stenosis were found to have abnormal metabolism of sphingomyelin, taurine/hypotaurine, pyrimidine, and protein (peptide) with the major involvement in taurine/hypotaurine, glycerophospholipid, and sphingolipid metabolism pathways [17]. Hydroxytetradecanoylcarnitine (C14OH) was detected to have positive correlation with carotid plague area after age adjustment [18]. The levels of metabolites related to the eicosanoid and beta-oxidation pathways were detected higher in symptomatic carotid plaque tissue than non-symptomatic one [19]. Nonenzymatic lipid peroxidation, mainly 9-hydroxyeicosatetraenoic acids, was abundant in advanced atherosclerosis and may promote plaque instability [20]. Nevertheless, there is no report using the targeted metabolomics in combination with machine learning to identify the metabolomics biomarkers in patients with severe extracranial CAS. The present study hypothesized that the use of targeted metabolomics in combination with machine learning techniques could help in identifying certain plasma metabolites that may be associated with the pathogenesis of CAS and may predict the presence of CAS.

Our contributions are as follows. We have (1) proposed a novel framework using decision tree and random forest techniques to predict the presence of CAS through targeted metabolomics methods; (2) compared the targeted metabolomics factors in different severity subgroups of CAS to predict the progression of atherosclerosis; (3) proposed a metabolomics framework to support the early prediction of CAS and raise the possibility of adjuvant metabolite therapy to prevent the progression of CAS to reduce the risk of stroke.

## 2. Materials and Methods

### 2.1. Subject Recruitment

Consecutive ischemic stroke patients with CAS were recruited from 2010 to 2015 according to the following inclusion criteria: (1) patients had at least one extracranial carotid artery ≥ 50% diameter stenosis in any segment from common carotid to extracranial internal/external carotid artery by cerebral angiography (digital subtraction, computed tomographic or magnetic resonance angiography) according to North American Symptomatic Carotid Endarterectomy Trial (NASCET) criteria [21], (2) blood samples were collected over 14 days after stroke onset with stable neurological condition, (3) patients had no acute illness at the time of blood sample collection such as infection or progressing stroke, and (4) modified Rankin scale ≤ 3. The normal controls were recruited in the same time period from a neurology outpatient clinic. Normal controls were defined as (1) no stroke and coronary artery disease (CAD) history, (2) carotid duplex, and brain magnetic resonance or computed tomographic angiography showed <50% diameter stenosis at bilateral intracranial and extracranial carotid arteries, and (3) no acute illness during blood sample collection such as infection. In all normal controls and CAS patients, the exclusion criteria included (1) female subjects, (2) having systemic diseases, such as hypo/hyperthyroidism, decompensated liver cirrhosis, acute kidney injury, or systemic lupus erythematosus, (3) presence of cancer and severe illness during the time of recruitment, and (4) normal controls with plaque score ≥ 1. This study was approved by the Institution Review Board, Linkou Chang Gung Memorial Hospital (number of revised approval document: 201506352B0C501 and 202000552B0C601). All subjects signed inform consent before the recruitment.

### 2.2. Carotid Plaque Score

All subjects had received carotid duplex for plaque score calculation. The accuracy of carotid duplex in the diagnosis of CAS was confirmed in our previous reports [9,22]. The time interval between carotid duplex and blood sample collection was within 30 days. Plaque score was evaluated according to our previous method [23] and was defined in the following degrees: 0 = diameter stenosis < 20%, 1 = 20–50%, 2 = 51–70%, and 3 = 71–100% in 12 segments, including bilateral common (proximal, middle, distal and bifurcation), internal and external carotid arteries. The summation of plaque score in the 12 carotid segments was used for analysis. Normal controls were defined as having plaque score = 0 in the 12 carotid segments. Patients with extracranial CAS was defined as having at least one carotid segment with plaque score ≥ 2. The distribution of plaque score in CAS patients and normal controls is presented in Appendix A. Age, clinical profiles, and laboratory blood tests were recorded for analysis.

### 2.3. Blood Sampling and Examination

Blood samples were collected at recruitment of normal controls and CAS patients in stationary condition. Blood for metabolomics was collected in sodium citrate tubes and then centrifuged immediately (10 min, 3000 rpm at 4 °C) within an hour after blood collection. Plasma was aliquoted into separate polypropylene tubes and immediately stored at −80 °C freezer. Measurement of other parameters, including homocysteine, high-sensitive C-reactive protein, lipid profiles, blood sugar, and kidney/liver function, was conducted at the Department of Laboratory Medicine in Linkou Chang Gung Memorial Hospital.

### 2.4. Metabolite Analysis

Metabolites were analyzed according to our previous method [24] with a commercial kit, the targeted Absolute IDQ^®^p180 kit (Biocrates Life Science, AG, Innsbruck, Austria). According to the manufacture manual, this kit contains a direct flow injection assay and a LC-MS/MS assay which can quantify a total of 194 endogenous metabolites from 5 classes of compound, including acylcarnitines, amino acids and biogenic amines, sugar, sphingomyelins, and glycerophospholipids. The LC-MS/MS assay was performed by using a Waters Acquity Xevo TQ-S instrument (Waters, Milford, MA, USA). First, the samples were thawed, vortexed, and centrifuged at 13,000× *g*. Then, a 10-uL aliquot of sample supernatant was loaded onto filter paper, dried under nitrogen flow, and derivatized by adding 20 µL of 5% phenyl-isothiocyanate for 20 min. Second, the filter spots were dried under nitrogen flow for 45 min. Then, the metabolites were extracted by adding 300 µL of methanol containing 5 mM ammonium acetate. Third, the extracts were analyzed by mass spectrometry by injection onto an Acquity UPLC BEH C18 (2.1 × 75 mm, 1.7-µm particle size, Waters, Milford, MA, USA) at 50 °C for the chromatographic separation of amino acids and biogenic amines. The process was followed under negative electrospray ionization and multiple reaction monitoring (MRM) mode and followed by flow injection analysis/thermospray mass spectrometry (FIA-MS)/MS of sphingolipids, hexoses, acylcarnitines, and glycerophospholipids. Finally, liquid chromatography data were quantified by performing with TargetLynx (Waters, Milford, MA, USA) based on an external 7-point calibration. The level of metabolite was obtained by converting and importing FIA data into the Biocrates^®^ MetIDQ™ software (10th version, BIOCRATES Life Sciences, AG, Innsbruck, Austria).

### 2.5. Statistical Analysis

The baseline characteristics and metabolite concentrations were presented as mean ± standard deviation (SD) for continuous variables with the statistics using Student’s *t* test and as count and percentage for categorical variables using chi-square test or Fisher’s exact test. To account for multiple testing, the Benjamini and Hochberg linear step-up method was adopted [25], and false discovery rate (FDR) adjusted *p* values (P_FDR_) were calculated using the MULTTEST procedure in SAS software (SAS Institute, Cary, NC, USA). A P_FDR_ value < 0.05 was considered statistically significant.

The metabolites were analyzed using principal components analysis (PCA) and orthogonal partial least squares discriminant analysis (OPLSDA) through the web-based metabolomics software MetaboAnalyst 5.0 (https://www.metaboanalyst.ca (accessed on 15 July 2022)). All metabolites were normalized by Pareto scaling. The variable importance in the projection (VIP) of each variable in the model was calculated to indicate its contribution to the classification. A higher VIP value indicates a stronger contribution to discrimination between groups. VIP values greater than 1.0 were considered significantly different.

The metabolites were further analyzed with decision tree and random forest analysis. The normal controls and CAS patients were randomly divided into training and testing groups in the ratio of 80% and 20%, respectively. The decision tree package (rpart) was used with the classification and regression tree (CART) method, and the model details were xval = 10, minsplit = 20, cp = 0.01, maxdepth = 30. RandomForest package (R software) was used for random forest analysis.

A total of 6 performance metrics, including accuracy, specificity, sensitivity, positive predictive value (PPV), negative predictive value (NPV), and area under the receiver operating characteristic (ROC) curve (AUC) were used to evaluate the performance of decision tree and random forest in the 3 subgroups of CAS patients. The split method was utilized to validate the results in which data were split into training and testing sets.

## 3. Results

### 3.1. Subject Characteristics

A total of 349 male subjects (176 normal controls and 173 ischemic stroke patients with ≥50% extracranial CAS) were consecutively recruited for metabolomics study (Figure 1). The distribution of plaque score in the 349 subjects is demonstrated in Appendix A. To further identify the normal controls with plaque score = 0 (127 subjects), 49 normal controls with plaque score ≥ 1 were excluded from the analysis (Figure 1). The three subgroups of CAS patients with plaque score ≥ 2, ≥5, and ≥8 were identified in 173, 142, and 79 CAS patients, respectively.

### 3.2. Baseline Characteristics

The baseline characteristic for normal control with plaque score = 0, and CAS patients with plaque score ≥ 2, ≥5, and ≥8 are presented in Table 1. Age, homocysteine, creatinine, and the percentage of comorbidity, including hypertension, diabetes, smoking, and alcohol consumption were significantly higher in the three subgroups of CAS patients when compared to normal controls (adjusted P_FDR_ value < 0.05). However, body weight, diastolic blood pressure, HDL-C, LDL-C, and total cholesterol were significantly lower in the three subgroups of CAS patients when compared to normal controls (adjusted P_FDR_ value < 0.05, Table 1). A comparison of metabolite levels between normal controls and the three subgroups of CAS patients is shown in Appendix A.

As the traditional metabolomics methods, PCA and PLSDA, did not show a clear separation between normal controls and the three subgroups of CAS patients (Appendix A), decision tree and random forest algorithms were employed for the construction of differentiation models.

### 3.3. Decision Tree Analysis

In decision tree analysis, the predictor metabolites and the cut-off points for each metabolite identified between plaque score ≥ 2 and 0 in the training group were demonstrated, as shown in Figure 2A. The root metabolite was butyrylcarnitine (C4). Subjects with C4 concentration ≥ 0.19 μmol/L could be divided into two groups according to tetradecenoylcarnitine (C14:1) concentration. The subjects with C14:1 concentration < 0.07 μmol/L mainly consisted of CAS patients (58 in 63, 92.1%). Subjects with C14:1 ≥ 0.07 μmol/L were then split into two groups according to symmetric dimethylarginine (SDMA). Subjects with SDMA concentration ≥ 0.75 μmol/L were all CAS patients (10 in 10, 100%). Subjects with SDMA < 0.75 μmol/L were further split into two groups according to phosphatidylcholine with acyl-alkyl residue sum C32:2 (PC ae C32:2). Subjects with PC ae C32:2 concentration ≥ 0.34 μmol/L mainly consisted of normal controls (10 in 11, 90.9%).

Subjects with C4 < 0.19 μmol/L were split into two groups according to phosphatidylcholine with diacyl residue sum C40:2 (PC aa C40:2). Subjects with PC aa C40:2 concentration < 0.21 μmol/L mainly consisted of CAS patients (27 in 33, 81.8%). Subjects with PC aa C40:2 ≥ 0.21 μmol/L were further split according to decenoylcarnitine (C10:1). Subjects with C10:1 concentration ≥ 0.39 μmol/L mainly consisted of normal controls (50 in 56, 89.3%). Subjects with C10:1 concentration < 0.39 μmol/L were then split according to kynurenine. Subjects with kynurenine concentration ≥ 1.79 μmol/L mainly consisted of CAS patients (19 in 24, 79.2%). Subjects with kynurenine concentration < 1.79 μmol/L were further split according to methionine. Subjects with methionine < 18.75 μmol/L mainly consisted of CAS patients (eight in 11, 72.7%). Subjects with methionine ≥ 18.75 μmol/L mainly consisted of normal controls (19 in 21, 90.4%).

The same decision tree analysis of plaque score ≥ 2 (Figure 2A) was also applied to differentiate the normal controls and patients with plaque score ≥ 5 (Figure 2B) and ≥ 8 (Figure 2C) to create another two models in the training group. As shown in Figure 2, C4 remained the root metabolite in the three plaque score subgroups. C14:1 and SAMA in plaque score ≥ 2 were the main metabolites, while octadecanoylcarnitine (C18) and phosphatidylcholine with diacyl residue sum C36:6 (PC aa C36:6) in plaque score ≥ 5 and phosphatidylcholine with acyl-alkyl residue sum C34:3 (PC ae C34:3) in plaque score ≥ 8 were the main metabolites in the differentiation between normal controls and CAS patients with a positive rate over 90% for CAS.

### 3.4. Random Forest Analysis

In the random forest analysis using the training group, the analysis of normal controls and CAS patients with plaque score ≥ 2 showed that seven metabolites had a mean decrease in Gini score ≥ 2.0, including C4, asparate, PC ae C34:3, C10:1, C14:1, C10, and proline (Figure 3A). The analysis of CAS patients with plaque score ≥ 5 showed only five metabolites (C4, C10, SDMA) with a mean decrease Gini score ≥ 2.0 (Figure 3B). However, in the analysis of CAS patients with plaque score ≥ 8, C4 had mean decrease gini score ≥ 2.0 (Figure 3C).

After creating the predictive models in the training group, the predictive models were applied to the testing group. The results of the six metrics in the evaluation of the performance of decision tree and random forest in the three subgroups of CAS patients among training, testing, and total subjects are shown in Table 2. In the analysis of training groups, the performance of the six metrics was greater than 0.8, except the specificity (0.782) in the decision tree of plaque score, which was ≥ 2. In the analysis of testing groups, the performance of the six metrics was greater than 0.6, except the PPV (0.588) in the decision tree of plaque score, ≥ 8. In the analysis of total subjects, the performance of the six metrics was greater than 0.7, except the specificity (0.642) in the decision tree of plaque score ≥ 2 and the sensitivity (0.647) in the decision tree of plaque score ≥ 8. The results of ROC curve and AUC are presented in Appendix A. The AUC results of training, testing, and total groups in plaque score ≥ 2, 5, 8 showed values greater than 0.7 in both decision tree and random forest (Table 2).

### 3.5. Metabolite Panel Analysis

For further evaluation, the metabolite panels which included the metabolites with a positive rate ≥ 90% for CAS in decision tree and the mean decrease Gini ≥ 1.0 in random forest (except C18 and PC aa C36:6) were used to evaluate the six performance metrics in the total of 349 subjects. The performance metrics showed that the detection of CAS had high specificity (0.923 ± 0.081, 0.906 ± 0.086 and 0.881 ± 0.109) but low sensitivity (0.230 ± 0.166, 0.240 ± 0.176 and 0.271 ± 0.169), with accuracy (0.580 ± 0.044, 0.635 ± 0.025, 0.743 ± 0.049) in different plaque score ≥ 2, ≥ 5, and ≥ 8, respectively (Table 3).

## 4. Discussion

Our study demonstrated that C4 is the root metabolite in the detection of CAS in the three subgroups. Acylcarnitine species (C4, C14:1, C18), amino acids and biogenic amines (SDMA), and glycerophospholipids (PC aa C36:6, PC ae C34:3) may be involved in the biomarker evaluation of CAS. The metabolite panels could be used as a more effective method than single metabolite to evaluate the possibility of CAS. Different metabolites may be involved in the varied severity of CAS.

Disturbances in lipid and carbohydrate metabolism, branched-chain and aromatic amino acid metabolism, as well as oxidative stress and inflammatory pathways observed on ^1^H NMR spectroscopy were found to have associations with subclinical atherosclerosis, which were consistent between coronary and carotid vascular beds [11]. In the studies of stable CAD, branched-chain amino acid metabolites and urea cycle metabolites were found to be associated with CAD [15], and reduced global arginine bioavailability ratio and increased citrulline levels were associated with an increased frequency of severe CAD [16]. A population-based Rotterdam study using ^1^H NMR found that location-specific metabolites, including glycolysis-related metabolites, lipoprotein subfractions, and amino acids, were associated with the etiology of intracranial and extracranial carotid artery calcification [12].

In the study of acylcarnitine, a previous report has shown the concentrations of acylcarnitines, including acetylcarnitine (C2), hydroxybutyrylcarnitine (C3DC), tetradecenoylcarnitine (C14:1), hydroxytetradecenoylcarnitine (C14OH), hydroxyhexadecanoylcarnitine (C16OH) and behenic carnitine (C22) had significant differences between male participants with and without carotid atherosclerosis [18]. However, after the adjustment for age, only C2 and C14OH were positively correlated with carotid plague area [18]. Aggregated short-chain acylcarnitines were reported as being associated with progression of carotid artery atherosclerosis in HIV-positive individuals, especially in those without persistent viral suppression [26]. Decreasing the acylcarnitine pools in apolipoprotein E knockout (apoE−/−) mice is able to attenuate the development of atherosclerosis [27].

C4 is found to be a good prognostic marker with an increase in hypoxic-ischemic encephalopathy and decrease after hypothermia treatment [28]. Three acylcarnitine species, including C4, hexanoylcarnitine (C6), and palmitoylcarnitine (C16), were detected with high intensity in carotid plaque samples of symptomatic stroke patients [19]. However, plasma C4 level was found being decreased in the atherosclerosis rats fed with high fat diet [29]. The compound including octadecanoylcarnitine (C18) proved positively correlated with the risk of diabetic cardiomyopathy [30].

In the male population, it has been reported that C14:1 measured by liquid chromatography mass spectrometry was positively correlated with age, but C2 and C14OH were positively correlated with carotid artery plague area after adjustment for age [18]. Elevated plasma levels of some acylcarnitine metabolites including C14:1 were found to be associated with cardiovascular disease risk, including stroke in type 2 diabetes [31]. The level of C14:1 was reported being similar between stroke patients and stroke-free controls [32]. However, our results showed that C14:1 was lower in extracranial CAS stroke patients compared to normal subjects, and the more severe the plaque score is, the higher the C14:1 level becomes (Appendix A). However, the remaining metabolites that were included in our metabolite panels had a similar level difference to the previous reports [32,33,34]. The reasons of different results of C14:1 levels in diseased subjects and controls may be related to the technique diversity, biofluid variation, different stroke populations, heterogeneous stroke etiologies, and various sampling times [35].

In the study of amino acids and biogenic amines, SDMA is reported as an endogenous inhibitor of nitric oxide synthase activity and is considered a novel risk factor for endothelial dysfunction and cardiovascular disease [36]. However, SDMA was not found to be significantly related to the cardiovascular disease risk factors in patients with rheumatoid arthritis [37]. In patients with cardiac diseases, SDMA was increased in patients with chronic renal failure [33]. A review article suggested that SDMA has the potential to represent a strong and reliable marker of vascular disease, renal dysfunction, and cardiovascular risk [38].

Phosphatidylcholine (PC) belongs to glycerophospholipid family and plays a structural role in cell membrane and blood lipoprotein [34]. In a systematic review, McGranaghan et al. discussed the metabolomic biomarkers for cardiovascular risk and found glycerophospholipids occupied the largest number of biomarkers reported in the literature [39]. Previous studies in the Atherosclerosis Risk in Communities study have found that lower concentration of PC aa C36:6 was associated with worse cognitive status and poor physical function in the elderly [40,41]. PC ae C34:3 was found to be correlated negatively with heart rate in patients with peripheral and coronary artery disease, suggesting its chronotropic effect which may act via cardiac ion channels [34]. A recent report also demonstrates that the modification of the glycerophospholipid metabolism pathway may help Herba patriniae to regulate lipid metabolism and inflammation for the treatment of atherosclerosis [42]. The alteration of specific pathways of glycerophospholipid and sphingolipid metabolism has been identified between the atherosclerosis-prone and age-matched atherosclerosis-resistant apoE−/− mice [43], suggesting that disturbances of metabolism in these lipids may be involved in atherosclerosis-related diseases [44,45].

The present study has the novelty of using LC-MS in combination with machine learning to define the metabolites that carry the potential to predict the presence of CAS and to define the components of metabolites that can help predicting the progression of CAS. The future perspectives are to use simple blood test of metabolites to early predict CAS and to prevent the progression of CAS by using adjuvant metabolite therapy.

There are also some limitations. First, we only studied the metabolomics in male subjects to avoid the influence of gender difference. It has been reported that males and females have some major differences in metabolomics components as well as phenotypic characteristics [46]. So, it is suggested that a separate statistical analysis should be considered carefully when detecting novel biomarkers and discovering a diagnostic algorithm for metabolic disorders to increase the statistical significance [46,47]. Second, our decision tree and random forest analysis did not include the clinical parameters. Table 1 shows that diastolic blood pressure, HDL-C, LDL-C, and total cholesterol were significantly lower in the three subgroups of CAS patients when compared to normal controls, which was likely due to the intensive medical controls in CAS patients. It has been reported that adjustment for conventional vascular risk factors may cause the attenuation of metabolite association [11] and result in inadequate results. Moreover, the metabolomics-based models can help in imputing conventional clinical variables when phenotypic variables are incomplete or unobserved in large epidemiological and clinical studies [48]. Third, the application of metabolic panels may need validation in another group of CAS patients. However, due to the limited number of CAS patients, further recruitment is needed.

## 5. Conclusions

Our previous genome-wide association study found two important genes being related to extracranial CAS, papillary thyroid carcinoma susceptibility candidate 3 (PTCSC3) and HDAC9 [6]. In association with the present metabolomics results, it is possible that there is a potential “metabolite–gene” regulatory axis to act on CAS, which may help to create a theoretical basis for the identification of CAS. It is likely that distinct plasma metabolomic biomarkers could be useful to monitor the development of CAS and help in differentiating the different stages of atherosclerotic progression. Diet adjustment to reduce the production of certain metabolites may be an adjuvant therapy to reduce the progression of CAS.

## Figures and Tables

**Figure 1 cells-11-03022-f001:**
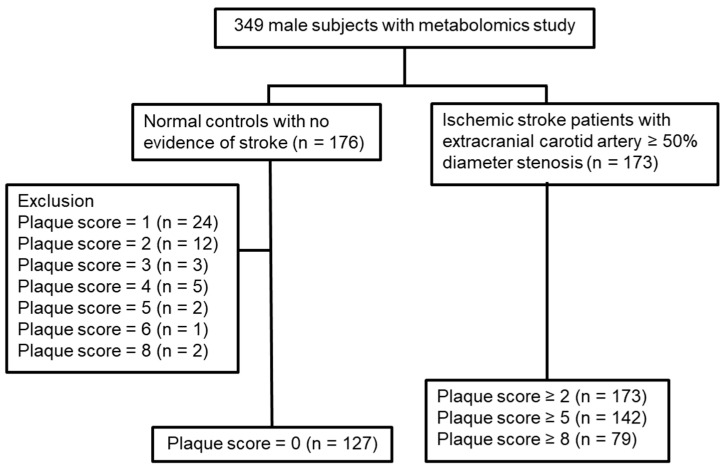
Flowchart of subject recruitment.

**Figure 2 cells-11-03022-f002:**
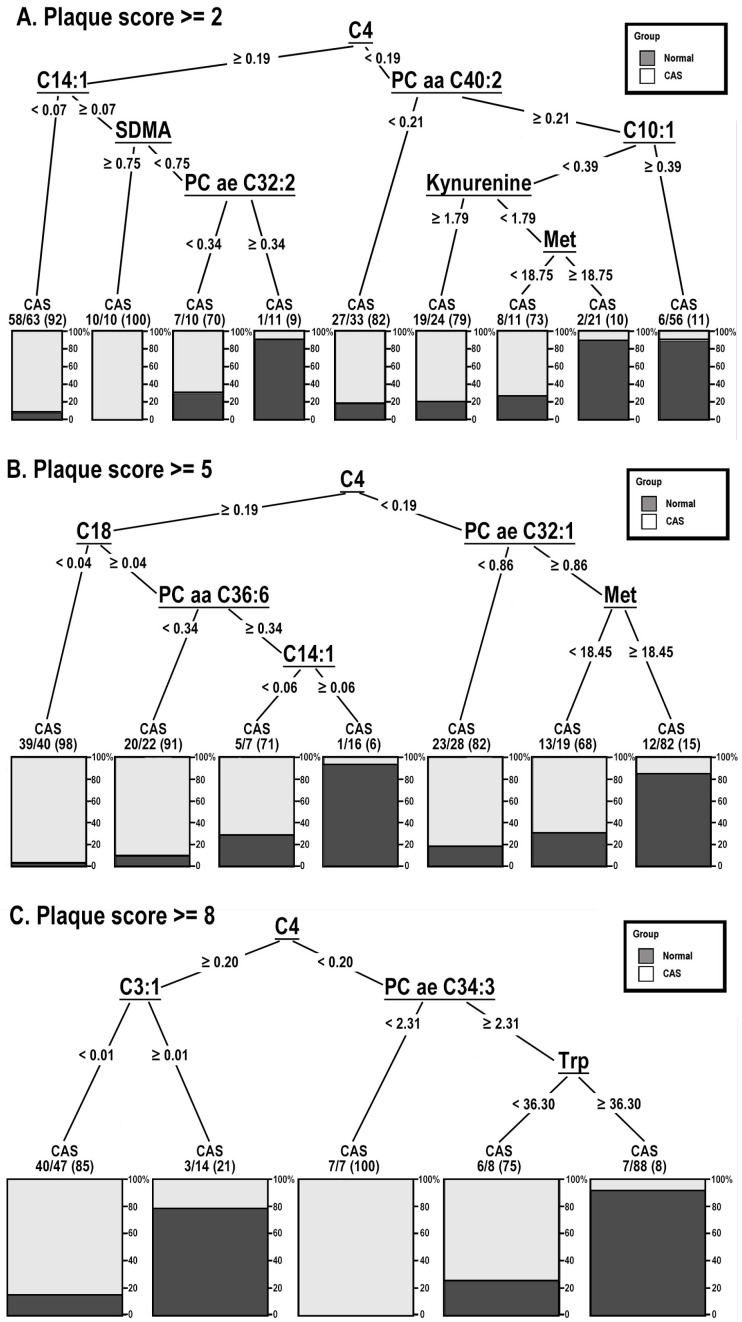
Decision tree study of normal controls (normal) and carotid artery stenosis (CAS) patients with plaque score ≥ 2, ≥5 and ≥8.

**Figure 3 cells-11-03022-f003:**
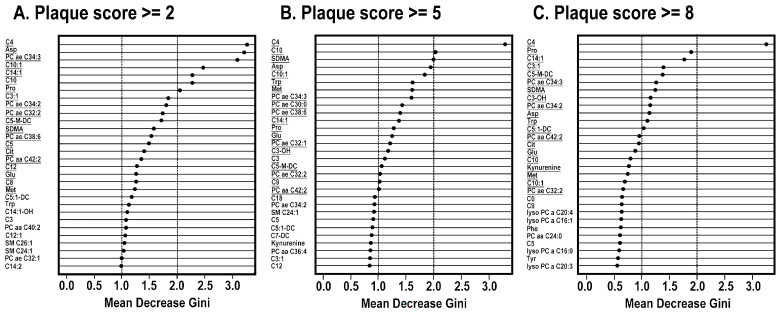
Random Forest study of normal controls (normal) and carotid artery stenosis (CAS) patients with plaque score ≥ 2, ≥ 5 and ≥ 8.

**Table 1 cells-11-03022-t001:** Comparison of baseline demographics between normal controls and the 3 subgroups of carotid artery stenosis.

Study Group	A. Plaque Score = 0	B. Plaque Score ≥ 2	Adj P_FDR_ A vs. B	C. Plaque Score ≥ 5	Adj P_FDR_ A vs. C	D. Plaque Score ≥ 8	Adj P_FDR_ A vs. D
Characteristics\No.	127	173		142		79	
Age	60.58 ± 6.23	64.19 ± 7.82	5.588 × 10^5^ *	65.00 ± 7.89	7.308 × 10^5^ *	66.38 ± 7.96	1.056 × 10^4^ *
Sex (Male)	127 (100)	173 (100)	0.581	142 (100)	0.738	79 (100)	0.971
Height	164.44 ± 7.80	163.44 ± 6.46	0.395	163.22 ± 6.65	0.309	162.92 ± 6.66	0.304
Weight	68.56 ± 10.38	65.50 ± 8.99	0.024 *	65.34 ± 9.07	0.026 *	64.13 ± 7.56	0.005 *
BMI	25.32 ± 3.28	24.52 ± 3.11	0.079	24.53 ± 3.17	0.125	24.18 ± 2.72	0.039 *
Waist circumference	85.01 ± 8.71	85.11 ± 8.89	0.941	85.00 ± 9.14	0.992	84.21 ± 8.49	0.669
Hip circumference	91.82 ± 6.36	89.98 ± 7.40	0.078	90.13 ± 7.61	0.143	89.72 ± 7.42	0.101
Systolic blood pressure	132.97 ± 18.45	135.88 ± 23.43	0.382	137.58 ± 24.34	0.170	139.35 ± 27.03	0.160
Diastolic blood pressure	81.60 ± 11.01	75.84 ± 13.30	5.588 × 10^5^ *	75.62 ± 13.30	0.001 *	74.32 ± 13.40	1.056 × 10^4^ *
Mean blood pressure	98.72 ± 12.36	95.86 ± 14.90	0.150	96.27 ± 15.31	0.286	96.00 ± 16.34	0.358
Heart rate	75.83 ± 13.70	74.37 ± 14.42	0.530	75.08 ± 14.92	0.778	76.01 ± 13.68	0.957
Homocysteine	10.17 ± 2.72	11.77 ± 3.95	5.588 × 10^5^ *	11.71 ± 4.12	0.004 *	12.14 ± 5.12	0.020 *
AC Sugar	98.31 ± 14.85	105.41 ± 30.28	0.031 *	105.07 ± 29.68	0.064	106.85 ± 33.19	0.101
HsCRP	2.65 ± 8.03	5.04 ± 12.57	0.121	5.21 ± 13.03	0.144	6.06 ± 15.98	0.220
HDL-C	49.39 ± 10.92	41.06 ± 10.42	5.588 × 10^5^ *	41.16 ± 10.66	7.308 × 10^5^ *	40.38 ± 8.91	1.056 × 10^4^ *
LDL-C	120.21 ± 39.92	104.89 ± 36.34	0.003 *	104.44 ± 36.05	0.004 *	106.09 ± 39.52	0.047 *
Triglyceride	130.44 ± 72.35	134.78 ± 75.11	0.724	136.60 ± 77.34	0.645	137.84 ± 82.44	0.638
Cholesterol	192.69 ± 35.06	172.45 ± 41.49	5.588 × 10^5^ *	172.54 ± 41.72	7.308 × 10^5^ *	173.58 ± 46.52	0.012 *
Uric acid	6.22 ± 1.38	6.35 ± 1.68	0.583	6.41 ± 1.68	0.452	6.34 ± 1.68	0.689
Creatinine	0.86 ± 0.16	1.04 ± 0.38	5.588 × 10^5^ *	1.05 ± 0.39	7.308 × 10^5^ *	1.02 ± 0.39	0.005 *
Comorbidity	No. (%)	No. (%)	*p* value	No. (%)	*p* value	No. (%)	*p* value
Hypertension	49 (39)	130 (75)	1.700 × 10^8^ *	107 (75)	1.010 × 10^7^ *	59 (75)	4.314 × 10^5^ *
Diabetes mellitus	1 (1)	55 (32)	1.900 × 10^9^ *	49 (35)	2.000 × 10^10^ *	34 (43)	<0.001 *
Smoking	50 (39)	128 (74)	1.031 × 10^7^ *	102 (72)	5.234 × 10^6^ *	59 (75)	5.017 × 10^5^ *
Alcohol	32 (25)	74 (43)	0.006 *	62 (44)	0.006 *	34 (43)	0.030 *
FHx stroke	41 (32)	64 (37)	0.536	52 (37)	0.601	31 (39)	0.469

Data are presented as mean ± SD for numerical data and number (%) for categorical data. Variables are analyzed by Student’s *t*-tests for numerical data or Chi-square test for categorical data. Model significance is presented in adjusted *p* value (Adj P_FDR_). * indicates *p* value < 0.05. AC, Ante cibum (before meals); BMI, body mass index; HDL-C, high-density lipoprotein-cholesterol; HsCRP, high-sensitive C-reactive protein; LDL-C, low density lipoprotein-cholesterol; FHx stroke, family history of stroke.

**Table 2 cells-11-03022-t002:** Performance metrics of decision tree and random forest in the 3 subgroups of carotid artery stenosis patients among training, testing and total subjects.

Plaque Score		Patient No.	Accuracy	Specificity	Sensitivity	PPV	NPV	AUC
Decision tree								
0 vs. ≥ 2	Training	239	0.870	**0.782**	0.935	0.854	0.898	0.897
	Testing	61	0.754	0.654	0.829	0.763	0.739	0.752
	Total	349	0.777	**0.642**	0.913	0.715	0.883	0.811
0 vs. ≥ 5	Training	214	0.864	0.842	0.885	0.862	0.867	0.903
	Testing	55	0.727	0.731	0.724	0.750	0.704	0.750
	Total	349	0.779	0.739	0.821	0.755	0.807	0.818
0 vs. ≥ 8	Training	164	0.884	0.911	0.8413	0.855	0.902	0.897
	Testing	42	0.690	0.731	0.625	**0.588**	0.760	0.714
	Total	349	0.751	0.852	**0.647**	0.812	0.711	0.754
Random forest								
0 vs. ≥ 2	Training	239	1.000	1.000	1.000	1.000	1.000	1.000
	Testing	61	0.770	0.654	0.857	0.769	0.773	0.881
	Total	349	0.877	0.784	0.971	0.816	0.965	0.979
0 vs. ≥ 5	Training	214	1.000	1.000	1.000	1.000	1.000	1.000
	Testing	55	0.855	0.846	0.862	0.862	0.846	0.933
	Total	349	0.874	0.841	0.908	0.849	0.902	0.966
0 vs. ≥ 8	Training	164	1.000	1.000	1.000	1.000	1.000	1.000
	Testing	42	0.857	0.923	0.750	0.857	0.857	0.887
	Total	349	0.851	0.938	0.763	0.923	0.801	0.923

Bold numbers indicate the lowest metric among training, testing and total subjects. Total 349 subjects = 176 normal controls + 173 patients with carotid artery stenosis; PPV: Positive predictive value; NPV: Negative predictive value; AUC: area under the receiver operating characteristic curve.

**Table 3 cells-11-03022-t003:** Performance metrics of metabolite panels in the 3 subgroups of carotid artery stenosis patients among the total 349 subjects.

Metabolite Panel	Plaque Score	Accuracy	Specificity	Sensitivity	PPV	NPV
C4 ≥ 0.19 + C14:1 < 0.07	≥2	0.645	0.852	0.434	0.743	0.605
	≥5	0.676	0.826	0.458	0.644	0.690
	≥8	0.702	0.767	0.481	0.376	0.835
C4 ≥ 0.19 + C14:1 ≥ 0.07 + SDMA ≥ 0.75	≥2	0.533	0.989	0.069	0.857	0.519
	≥5	0.622	0.990	0.085	0.857	0.612
	≥8	0.791	0.985	0.127	0.714	0.794
C4 ≥ 0.19 + C18 < 0.04	≥2	0.602	0.818	0.382	0.673	0.574
	≥5	0.639	0.802	0.401	0.582	0.661
	≥8	0.682	0.759	0.418	0.337	0.817
C4≥ 0.19 + C18 ≥ 0.04 + PC aa C36:6 < 0.34	≥2	0.562	0.972	0.145	0.833	0.536
	≥5	0.628	0.957	0.148	0.700	0.621
	≥8	0.785	0.952	0.215	0.567	0.806
C4 < 0.2 + PC ae C34:3 < 2.31	≥2	0.556	0.983	0.121	0.875	0.532
	≥5	0.610	0.957	0.106	0.625	0.609
	≥8	0.756	0.944	0.114	0.375	0.785
Mean ± standard deviation	≥2	0.58 ± 0.04	0.92 ± 0.08	0.23 ± 0.17	0.80 ± 0.09	0.55 ± 0.04
	≥5	0.64 ± 0.03	0.91 ± 0.09	0.24 ± 0.18	0.68 ± 0.11	0.64 ± 0.04
	≥8	0.74 ± 0.05	0.88 ± 0.11	0.27 ± 0.17	0.47 ± 0.16	0.81 ± 0.02

PPV: Positive predictive value; NPV: Negative predictive value.

## Data Availability

The data presented in this study are available on request from the corresponding author. The data are not publicly available due to privacy.

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
