# Peer review of "Identification of Metabolomics Biomarkers in Extracranial Carotid Artery Stenosis"

_cells, 2022, doi:10.3390/cells11193022_

Round 1

Reviewer 1 Report

The manuscript ” Identification of metabolomics biomarkers in extracranial carotid artery stenosis” aimed to identify specific metabolomic biomarkers for prognosis of carotid artery stenosis. Metabolomic is a promising tool used in disease diagnosis and progression evaluation and its used in stroke is currently limited. Due to increased mortalidy determined by carotid artery stenosis, the identification of new biomarkers are eminent for the evaluation of the disease. In this study the authors aimed to identify new targeted metabolomics biomarkers associated with the pathogenesis of carotid artery stenosis that can help in disease prevention and management. The idea of the study is of novelty and great interest for the research community. I have the following comments for the authors:

1.       Define all the abbreviation at the first use in the text, eg: NASCET criteria, line 67, etc.

2.        Please clarify why you excluded females from the study

3.       I suggest in table 1 to present with * the parameters that present statistically significant differences compared with the control group to can have a more easy and clear image of the results.

4.       In the discussion section add the novelty of the results obtained and the future perspectives. 

Reviewer 2 Report

The paper will be accepted if the following comments are incorporated well.

1.      In the abstract “The metabolomics biomarker related to carotid artery stenosis (CAS) has not been studied”. This is a hard statement. Kindly prove the statement with scientific evidence.

2.      Abstract needs to be more technical.

3.      The contribution is unclear. Refer to the following paper, and check how to specifically write the contribution at the end of the Introduction section. “A novel framework for prognostic factors identification of malignant mesothelioma through association rule mining”.

4.      Introduction section is short. In the introduction, the background of the research domain, the work of existing research, and the difference between the authors’ work and existing research should be stated clearly.

5.      Clearly mention the research rap/rationale using the latest research. Furthermore, add a separate section of related work/literature review and cite the papers regarding biomarkers identification as mentioned: Metabolomics study in severe extracranial carotid artery stenosis; A machine learning approach for identification of malignant mesothelioma etiological factors in an imbalanced dataset; Metabolomics and lipidomics profiling in asymptomatic severe intracranial arterial stenosis: results from a population-based study; Early Prediction of Malignant Mesothelioma: An Approach towards Non-invasive Method; Risk factors identification of malignant mesothelioma: A data mining based approach;  

6.      The inclusion criteria has explained well but exclusion criteria are not discussed.

7.      Compare the results of decision tree variations other than random forest and report their results.

8.      Another evaluation measure (ROC) must be computed and report the results.

9.      Paper needs a threat to validity section refers to the following paper “An Investigation of Credit Card Default Prediction in the Imbalanced Datasets”.

10.  Authors should use some hypothesis testing to prove or validate the results.

11.  Add the future directions and limitations of the study in the conclusion.

Overall speaking, the innovation points and main contributions of this paper need to be carefully reconsidered, and the innovation points should be presented more clear and prominent in terms of word expression and methodology & experiment design.

Round 2

Reviewer 2 Report

Accepted.